# Sportsmen’s Attitude towards Dietary Supplements and Nutrition Knowledge: An Investigation in Selected Roman Area Gyms

**DOI:** 10.3390/nu14050945

**Published:** 2022-02-23

**Authors:** Alberto Finamore, Luca Benvenuti, Alberto De Santis, Serena Cinti, Laura Rossi

**Affiliations:** 1Council for Agricultural Research and Economics—Research Centre for Food and Nutrition (CREA—Food and Nutrition), Via Ardeatina 546, 00178 Rome, Italy; alberto.finamore@crea.gov.it (A.F.); cintiserena@yahoo.it (S.C.); 2Department of Computer, Automation and Management Engineering, Sapienza University of Rome, Via Ariosto 25, 00185 Rome, Italy; luca.benvenuti@uniroma1.it (L.B.); alberto.desantis@uniroma1.it (A.D.S.)

**Keywords:** sport, nutrition knowledge, dietary supplements, non-professional sportsmen, gyms

## Abstract

The non-professional sport environment is a grey zone not as widely assessed as that of elite athletes. The purpose of this research was to investigate the dietary supplementation habits and the nutrition knowledge on sport (NKS) in a sample of gym users. The level of adequacy of NKS was set at ≥60% of correct answers. Almost half (46.4%) of respondents stated they used food supplements, in particular multivitamins (31.0%), amino acid pills (29.5%), minerals (29.1%), and protein powders (28.7%). Supplements were used to increase muscle mass (36.9%) and to repair muscle (35.1%). Gym trainers were the preferred source of information on the use of supplements, especially in males (84%). The NKS correct response rate was 57.1% and the proportion of respondents with a sufficient level of NKS was 47.3%. The prevalence of correct answers was highest in males (61.5%) and for respondents with the highest educational attainment levels (44.5% and 53%). This study demonstrated that non-professional sportsmen do not have sufficient knowledge of nutrition and that the gym environment does not facilitate the circulation of the correct information on the role of supplementation. Considering the importance of nutrition for sportsmen, it is necessary to put in place actions aimed at increasing the knowledge of nutrition of gym users and their trainers.

## 1. Introduction

A balanced diet and regular physical exercise are essential for good health. Sport, physical exercise, and an active lifestyle, along with healthy food consumption patterns, are fundamental to maintaining a state of well-being by reducing the risk of obesity and the occurrence of other comorbidities [1,2]. In Western societies, gym centers are regularly frequented in order to maintain or improve functional capacity and physical appearance. Regular physical activity increases energy and nutrient requirements; thus, nutrition plays a key role in the improvement of performance, training adaptation, renewal of energy stores, and reduction of fatigue and recovery time [3]. Adequate training and balanced nutrition are essential for sport; in addition, several pieces of evidence demonstrate that the appropriate intake of selected nutritional supplements can enhance performance [4]. Dietary supplements contain a wide variety of ingredients including macro and micronutrients (e.g., minerals, vitamins, proteins, amino acids) and ergogenic supplements (e.g., creatine, caffeine, β-alanine), among others. Dietary supplements are commercialized in different pharmaceutical forms including pills, beverages, bars, gels, liquid meals, etc. It has been found that dietary supplements are widely used among athletes and non-competitive sportsmen and women, depending on the type of sport and the level of training and competition [5]. Some studies have highlighted that dietary supplements are widely used by non-competitive gym participants for various reasons, such as performance amelioration, weight loss, muscle gain, and improvements of physical fitness for aesthetic reasons [6,7,8]. The use of these substances in non-competitive gym users may not lead to real benefits and could be harmful. Thorough knowledge of these products is fundamental to avoid misuse and to ensure understanding of what a supplement and its correct use is. Information concerning supplements is acquired through social media, magazines, protein shops, the internet, or from untrained personnel [8]. Usually, the main sources of information or recommendations about supplements are coaches and instructors, even if they are not always sufficiently knowledgeable about the relationship between nutrition and sport. Furthermore, supplements can be purchased easily through various suppliers, such as protein shops and gym shops, as well as online suppliers; this easy access to dietary supplements increases the likelihood of consumption by non-professional gym users [9]. Regulations on supplements differ across countries and there are supplements that can be found in illegal markets that are categorized as unsafe and that do not meet required quality standards. These products could be contaminated by harmful and prohibited substances that are undeclared, such as heavy metals and anabolic steroids, which may represent a health risk for users [2,5,10,11,12]. Several studies have also indicated adverse effects associated with dietary supplement consumption, especially if the intake goes beyond recommended quantities; they include cardiovascular, hematological, metabolic, and neurological problems [13,14].

In Italy, if a product is categorized as a dietary supplement, it does not require a medical prescription and is easily available. Often these products are selected based solely on opinion or belief. The main European Union legislation concerning food supplements is Directive 2002/46/EC [15]. To ensure an adequate level of quality and safety of dietary supplements, the Italian Ministry of Health published “Recommendations on the good manufacturing practices of food supplements” in 2018 to provide technical indications necessary to meet the specific obligations of the manufacturing industries [16].

According to Maughan et al. (2018) [4], nutritional supplements can be classified as dietary supplements, sport nutrition products, or ergogenic supplements. The category of dietary supplements mainly includes micronutrient supplements, such as vitamins and minerals, but also essential fatty acids. The main objective of supplementation is the compensation of nutrient deficiencies due to inadequate intake and/or increased need. Macronutrients, namely carbohydrates, proteins, and fats are contained in sport nutrition products, such as sports drinks, recovery drinks, energy bars, etc., while ergogenic supplements, including caffeine, beta-alanine, carbohydrates, sodium nitrate, and creatine, are products that are claimed to have performance-enhancing properties.

A previous study conducted in Italy by Cannataro et al., (2019) [17], with more than 3000 participants, showed that the supplements mostly used by sportsmen and women were whey protein, branch chain amino acids (BCAA), creatine, multivitamins, and “pre-workout” supplements. The main purpose for dietary supplementation was muscle building in males, weight loss in females, and health benefits for both sexes. The main sources of information on dietary supplements were social media, friends and training mates, trainers, and coaches, while only 4% of participants consulted qualified professional figures [17]. The types and amounts of supplements used by sportsmen change according to which sports they engage in, and the specialties associated with each. Bodybuilders, who pursue muscle hypertrophy, are keener to use dietary supplements and hormones than other gym users [18]. In sports that need sprint performance and high-intensity exercises, caffeine, beta-alanine, and other ergogenic supplements are used with the purpose of reducing fatigue and the time for muscular recovery, whereas creatine is normally used to improve performance during short and prolonged intermittent exercise. Recently, nitrate consumption has also been considered to have beneficial effects in high-intensity training [19].

According to Trakman et al. (2016) [20], nutrition knowledge can influence dietary choices and impact athletic performance; however, nutrition knowledge could depend on an athletic level, as it is plausible that elite athletes have greater access to resources and, therefore, higher levels of knowledge. Above all, it seems that sportsmen’s nutrition knowledge is related to what coaches consider the best for the athletes. Several studies [21,22,23,24] have reported the poor nutritional knowledge of athletes, as well as the fact that their primary sources of nutrition information are social media, coaches, and athletic trainers.

The category of non-professional gym users is not widely assessed, and, in Italy, few studies are available on the use of dietary supplements and nutrition knowledge related to sport in physically active subjects. In sportsmen, the idea of achieving the highest performance and the maximization of personal goals is prioritized more than the concept of psycho-physical well-being [25]. The purpose of this study was to investigate the dietary supplementation habits and the sports nutrition knowledge (NKS) level of people frequenting selected fitness centers in Rome and surrounding areas. Considering this aim, the hypothesis of this study was that, in a constantly evolving environment, such as fitness centers, in which nutrition aspects are increasingly popular and are often addressed by personal trainers without a specific nutrition background, the use of supplements, as well as the spreading of misinformation, would be common. The research questions underlying this study were: (i) What are the drivers that led participants to search for supplementation? (ii) To what extent does the inadequate NKS of this group of sportsmen affect their behavior? (iii) To what extent is it possible to intervene to fill NKS gaps at the level of the sports club?

## 2. Materials and Methods

### 2.1. Setting and Study Methodology

The present assessment is a cross-sectional study conducted on nonprofessional sportsmen who frequented selected fitness centers located in Rome and the surrounding province. The inclusion criteria were for participants to be aged between 20 and 50, to be regular attendants of sports centers, and to be engaged in different sports activities. Five sports centers within Rome and the surrounding areas were involved in the data collection.

The sample size was calculated according to Pourhoseingholi et al. (2013) [26], using the following formula
*n* = [Z^2^ ∗ P(1-P)]**/[**d^2^]
where n is the sample size, the Z value is 1.96 corresponding to a 95% level of confidence, P is the expected prevalence of dietary supplement use in the gyms, which was estimated at 30% according to Giammarioli et al., (2016) [27], Morrison et al., 2004 [28], and Del Balzo et al., (2014) [29] and expressed as a decimal; d is the precision level of 4%, expressed as a decimal. The data was calculated with a sample size of 504 subjects. This figure was incremented by 15% to compensate for data cleaning and missing values.

Data collection was performed between February and June 2017.

### 2.2. Data Collection Procedure

In order to obtain informed written consent from those who expressed willingness to participate, a nutrition expert described to the manager of the facility and to the attendants of the sports centers the objectives and procedures of the study. In accordance with the European Commission General Data Protection Regulation (679/2016), those willing to participate signed a privacy policy and consent form concerning the collection and processing of personal data in advance. Before starting data collection, participants were informed about the objectives of the research, the consequent statistical analysis, and the intention to publish the results of the assessment in a scientific journal. Participation in the study was fully voluntary and anonymous and subjects could withdraw from the study at any time, for any reason. This study was conducted according to the guidelines of the Declaration of Helsinki [30]. The assessment did not involve any invasive procedures or induce any change in dietary patterns. Therefore, the study did not require approval by the Ethics Committee.

### 2.3. The Questionnaire

The subjects who agreed to participate in the study were asked to complete a questionnaire developed from a combination of two previously validated questionnaires. To investigate socio-demographic aspects, the levels of physical activity performed, and the use of supplements, the questionnaire of Khoury and Antoine-Jonville (2012) [31] was used. The questionnaire used to investigate nutrition knowledge in sports was that developed by Horvath et al. (2014) [32]. The resulting questionnaire, reported in the Appendix A (The questionnaire), included 52 questions divided into four sections:

The first section included questions on demographic characteristics (e.g., age, sex, level of education), the occurrence of the previous disease, and lifestyle information (e.g., alcohol consumption and smoking).

The second section assessed the characteristics of the sporting activities (e.g., the type, the frequency, and training session duration), the occurrence of nutritional counseling, and the professionals consulted.

The third section investigated the use of supplements, the source of nutrition information, the type of nutritional supplement used, the reasons for consumption, and the duration and timing of intake.

The fourth section consisted of 33 questions aimed at assessing nutritional knowledge related to sport (NKS) using the questionnaire of Horvarth et al. [32], covering five topics: macronutrients (NKS1—9 questions), micronutrients (NKS2—6 questions), hydration (NKS3—7 questions), energy balance (NKS4—6 questions), and supplements (NKS5—5 questions).

The answering system consisted of closed-ended, multiple-choice, and yes/no questions. For the NKS section, participants indicated their responses to the proposed statements by answering “True”, “False”, or “Don’t Know”. Statements answered correctly were given a score of 1, and statements answered incorrectly, including those with the answer “Don’t Know”, were scored as 0, giving an overall score, referred to as the NKS score, of between 0–33. The NKS score was calculated for each participant by adding the total number of correct answers and calculating the percentage compared to the maximum expected score [e.g., (NKS score/33) ∗ 100]. A cut-off point of ≥60% of questions answered correctly was used to indicate adequate NKS [33].

### 2.4. Statistical Analysis

Descriptive statistics of the data collected were produced. Single continuous and categorical variables were summarized as mean ± standard deviation and percentage (%). A contingency analysis was performed to check associations between variables. Specifically, double-entry tables were processed, and the Chi-square test of independence was applied, along with post hoc tests to check pairwise comparisons, with Bonferroni corrections of the *p*-values. A *p*-value of less than 0.05 was indicative of statistical significance. The statistical analysis was performed using the IBM SPSS Statistics package, version 25.

## 3. Results

### 3.1. Sample Characteristics

Table 1 reports the sociodemographic and physical characteristics of the sample. The sample was mainly composed of young (50.3% in the range of 20–30 years) males (56.5%) with a medium level of education (52.8%). BMI, calculated based on self-reported weight and height, was mainly in the normal range (69.4%), with a proportion of respondents (25%) falling within the overweight range.

Most of the sample (91.4%) reported not having suffered pathologies, the remaining 8.6%reported occurrence of pathologies without specifying the typology. A total of 62.7% of participants reported alcoholic beverage consumption and 69.5% stated they were non-smokers (data not shown).

### 3.2. Typology of Sports and Diet Attitude

As shown in Table 2, most participants in the sample had taken part in regular physical activity for more than 1 year (78.5%), with a frequency of 3–5 times per week (59.7%), and a duration of 1–2 h per day (76.6%). Almost two-thirds of the sample (71.4%) carried out strength training. Approximately one-third (31%) of the sample followed a diet specifically designed for sport, with 48.3% having consulted a nutritionist, 24.4% having a self-prescribed diet, and in 16.1% of cases, following the advice provided by the gym’s trainer.

### 3.3. Attitude toward Food Supplements for Sport

Almost half (46.4%) of the respondents stated they used food supplements for sport. In the studied sample, food supplement users were most frequently (*p* < 0.05) male (61.9%), young (48.9% between 20 and 30 years old) and had been supplement users for more than 1 year (4.1%).

Figure 1 shows the type of food supplement used by the respondents. The most used food supplements were multivitamins (31.0%), amino acid pills (29.5%), minerals (29.1%), and protein powders (28.7%). Herbal supplement utilization was not reported.

As shown in Table 3, a statistically significant gender difference was observed concerning the pattern of food supplement selection, with males (61.9%) using more supplements than females (38.1%) (*p* < 0.05). There were also significant differences between sexes (statistically significant *p* < 0.001) concerning the use of amino acid pills (89.9%), creatine (83.2%), branched-chain amino acids (84.8%), protein powders (81.8%), carnitine (85.7%), and sport energy drinks (78.6%), which were more common in males than in females. Although the youngest respondents used more supplements than the oldest, there were not any significant differences. However, with respect to the products consumed by participants, statistically significant differences (*p* < 0.05) were found for the use of whey proteins (58.1%), group B vitamins (44.2%), and calcium (44.6%), consumption which was higher in young people (20–30 years old) than in other age categories. Conversely, iron supplements were used more frequently (statistically significant *p* < 0.003) by the oldest respondents (50% of participants 41–50 years old) compared to other age groups.

The reasons for using sport food supplements are reported in Figure 2. Approximately one-third of the sample used supplements to increase the muscular mass (36.9%) and to repair and enable recovery of muscle (35.1%), one-fourth of the sample declared using supplements to reduce post-workout fatigue (27.6%) and to increase muscular strength (25.4%), while increasing performance (16.8%) was less frequently reported. None of the respondents selected the option “Increase alertness and mental activity” as a reason for using sports supplements.

A well-defined gender pattern was observed in the reported rationale behind sports supplementation (Table 4). Males used food supplements to increase muscle mass and strength, for muscle recovery, to increase performance, to reduce fat mass, and to reduce post-workout fatigue; these were the reasons for which statistically significant differences were observed. For females, the only statistically significant result was supplementation for medical reasons. The items with statistical significance (*p* < 0.01) in terms of age categories (Table 4), were “increasing muscular mass”, being higher in the youngest age group (64.6% in 20–30 years old) than in the oldest, while “stress reduction” was higher in the oldest (40.6% in 41–50 years old) than in other age groups.

Figure 3 shows the supply modalities of the sub-sample that reported the use of sports supplements. Gym trainers (27.9%), nutritionists (27.1%), and medical doctors (21.9%) were the most reported professionals that prescribed food supplements in the present sample (Figure 3—Panel A). Almost one-third of respondents reported buying supplements from specialized sports nutrition stores (36.6%), in pharmacies (31.7%), and online (27.1%). The remaining percentage (4.6%) of the sample obtained supplements directly from trainers (Figure 3—Panel B).

Gym trainers were the preferred source of information for supplement prescription; with statistically significant differences (*p* < 0.001) found between sexes; males reported consulting gym trainers more frequently (84%) than females. No other significant differences were observed. Reading the labels of sport food supplements was reported by a large proportion of respondents (85.1%) (data not shown).

### 3.4. Multivariate Statistical Correlation Analysis

The use of supplements for sport was found to be correlated with only three characteristics among the physical and lifestyle factors (Table 1) and those related to the typology and timing of sports and diet attitude (Table 2). Multivariate correlation analysis was performed using a logistic regression model that expressed the natural logarithm of the odds of an event (e.g., the respondent uses supplements for sport) as a linear combination of the characteristics of the sample. A total of 87.1% of respondents who performed strength training, and followed a diet prescribed by a gym’s trainer specifically designed for sport, used supplements. The food supplements most frequently used by this sub-group were proteins and protein bars (77.8%), branched-chain amino acids (63.0%), amino acid pills (59.3%), and caffeine (47.6%).

In addition, supplement use was reported by 63.3% of respondents following a regimen for sport. The data showed that 84.4% of those following a diet prescribed by a gym trainer, and 52.8% of those performing strength training, were using supplements. The same analysis was performed on the characteristics of the respondents following a specific diet for sport. In this case, only five characteristics were found to be related to the pursuit of a diet. A total of 56.3% of respondents with BMI > 30 kg/m^2^, and 50.0% of those whose highest level of education was secondary school, followed a diet. In addition, most of the attendees who consumed alcoholic beverages (73.3%) did not follow a sport-specific diet. The same was true for 63.3% of attendees training for strength, and 66.3% of those training for fitness.

### 3.5. Nutritional Knowledge Related to Sport

The mean NKS score assessed in the present sample was 18.8 ± 5.4 with a correct response rate of 57.1% (Table 5). The proportion of respondents that had a sufficient level of NKS (meaning ≥60% of correct answers) was 47.3%. The theoretical maximum (33 points) was not achieved by any of the respondents, although there were differences among sections. NKS4, the energy balance section, achieved the highest rates (65.6% of correct answers and 64.2% of sufficient level of NKS), while the lowest percentages were found in the NKS5-Supplements section (41.2% of correct answers and 34.3% of sufficient level of NKS).

The NKS means were not statistically different by gender, age category, or BMI (data not shown).

The prevalence of correct answers was significantly higher in males (61.5%) than in females (38.5%), and in respondents with the highest educational attainment levels (44.5% and 53%). This prevalence did not differ based on the variables of age or BMI (Table 6). It is interesting to note that a statistically significant proportion of respondents with an insufficient level of NKS reported not using supplements (58.5%).

## 4. Discussion

The purpose of this study was to record the dietary supplementation habits and the NKS level of a sample of people who frequented fitness centers. Dietary supplementation was a common habit in the present sample, especially in young males; the reasons for supplementation were mainly related to increased muscle mass and strength. Overall poor NKS proficiency was found, with males who had achieved a higher level of education being the best informed in terms of NKS.

These data are of particular interest because, although there is much information on the use of dietary supplements in athletes [34], little data is available on the use of supplements in people that attend gyms for nonprofessional reasons. The findings of this study are in line with the results of Ruano and Teixeira (2020) [6], who reported that gym users are large consumers of dietary supplements, prevalent users are young and male, they use protein powders to increase muscle mass, consider themselves well informed, and buy supplements online. In a study carried out in Brazil, it was found that supplement intake in people exercising in gyms was high (36.8%), was usually self-prescribed, and the products most consumed were those rich in proteins and amino acids (58%) [35].

The specific use of dietary sports supplements for increasing muscular mass is confirmed by other studies carried out in Italy. According to Cannataro et al. (2019) [17], the main purpose of dietary supplementation was to build muscle in males, lose weight in females, and achieve health benefits for both sexes. In another Italian study, it was shown that dietary supplements are widely used by those involved in body shaping-oriented fitness training and that the gym users decided individually which supplement to use [10].

Unexpectedly, in this study, multivitamins were the most-reported supplement used by the respondents. This result did not match with the most reported reason for supplementation (i.e., increasing muscle mass) which would suggest that amino acids and proteins (in the second and fourth positions) would be the preferred options. The selection of multivitamins and minerals by those frequenting gyms could be related to the old concept that vitamins and minerals have ergogenic benefits for sport. Depending upon the focus of the sport, e.g., strength, speed, power, endurance, or fine motor control, athletes use megadoses of various vitamins in an attempt to increase specific metabolic processes important to improved performance [36]. However, while it is true that deficiencies in vitamins and minerals could impair the physical performance, once a deficiency is corrected, performance usually improves. Moreover, additional vitamin and mineral supplementation has not been shown to further improve performance in athletes with a well-balanced diet [5]—this is even more true for non-professional sports performers who do not have special micronutrient requirements. Where there is a demonstrated deficiency of an essential nutrient, increased intake from food or supplementation may help, but many sportsmen ignore the need for caution in supplement use and the harm that taking supplements in unnecessary doses could cause [37].

The present sample was characterized by the fact that, for them, gym trainers represented a reference source for prescription of supplements (27.9%), especially for males (84%). This is an important point to consider because gym trainers’ competence in food supplementation for sport is often based on anecdotical knowledge [2]. This finding could, in part, explain the large use of supplements in general, and of vitamins and minerals in particular, found in the present study. This is probably due to a misunderstanding of the role of vitamins and minerals in the diet, their function in maintaining overall health, their role in athletic performance, and how they are best obtained from the diet [38]. The role of coaches as references for supplement use in gym frequenters is reported in several studies. According to Nieper (2005) [39], coaches (65%) had the greatest influence on supplementation practices, with doctors (25%) and sports dieticians (30%) being less important. Wiens et al. (2014) [40] reported that the main sources of information were family/friends (74%), coaches (44%), athletic trainers (40%), medical doctors (33%), with sports nutritionists (32%) being less often cited.

According to Garthe and Maughan (2018) [5], there is some evidence—largely anecdotal but supported by some findings from surveys—that the quantity of supplements used by sportsmen often exceeds the recommended amount, following the philosophy that “more is better”, especially in a competitive setting, such as the gym environment. This attitude is of particular concern in situations, as assessed in this study, that involve a sample of people not professionally engaged in sports activities who do not have the opportunity to consult sport professionals for appropriate nutrition advice.

The general approach to sports supplementation found in this sample corresponded to the NKS level, resulting in a correct response rate of 57.1%, with only 47.3% of respondents having a sufficient level of NKS, and the theoretical maximum of correct answers never being achieved by respondents. The inadequacy of NKS in sportsmen has been reported in several other studies [41,42,43,44]. This is a key public health problem that needs to be addressed. Considering the importance of nutrition for people frequenting gyms, it is necessary to put in place actions designed to increase the knowledge of nutrition of sportsmen and their trainers.

In the present sample, males achieved higher NKS levels than females. When assessing general nutritional knowledge, meaning nutrition literacy not focused on sport, there were significant independent effects of gender on knowledge score, with males having poorer NKS than females [45,46]; in Italy, this was also confirmed in [47,48]. When the topic is nutrition knowledge in sport, the gender pattern is less clearly defined. Boumosleh et al. (2021) [41] showed that being male is associated with inadequate NKS in athletes. This was supported by Smith-Rockwell et al. (2001) [49] who reported that NKS levels of females were higher than those of males. However, in Ozdogan et al. (2011) [42], it was found that the mean NKS scores of male students were higher than those of female students. In Rosenbloom et al. (2002) [50], it was reported that NKS did not vary according to gender. In this sample, practicing sport is a driver for increased interest in nutrition for Italian males who normally do not pay attention to nutritional issues.

The relationship of NKS with the educational level found in this sample was expected and is confirmed by other studies [51,52]. One of the most relevant findings was the statistically significant proportion of respondents with an insufficient level of NKS who reported not using supplements (58.5%). A study carried out by Wardenaar et al. (2017) [53] evaluated the effect of dietary counseling on supplement use and found that athletes who had counseling used more supplements than those who did not. The reason for increased use was mainly due to recommendations for increased vitamin supplementation in elite athletes to enhance performance. In the present sample, it appeared that when more information on nutrition was provided, instead of increasing awareness and limiting the use of supplementation, it created the opposite situation, with a superficial knowledge in some aspects of sports and nutrition leading people to feel confident in deciding their personal nutritional strategies for sport. It seems that having a superficial level of knowledge resulted in the presence of beliefs that negatively influenced subjects’ behaviors more than being completely ignorant. This is an inference that would need to be analyzed in greater detail and more thoroughly if confirmed.

This study has several strengths, the most important relates to the fact that the participants of this study were part of a population group that has not been widely assessed. It is most common to find studies on elite athletes or sport professionals, but less frequent is the assessment of non-professional gym users, as carried out in the present study. The sample size enabled the identification of different parameters and their association after data gathering. The questionnaires were not emailed or sent by an online system, but instead were completed during face-to-face interviews, thus optimizing data quality.

The general limitation of this kind of study is linked to the use of self-reported answers that could affect the reliability of the responses and the fact that the measured variables were based on participants’ perceptions which may not reflect the real conditions. However, the quality of the data collected is supported by similar results in other, closely related, surveys. Another relevant limitation is that the use of these supplements may be difficult to admit in front of a nutritionist, thus affecting the data collected. Weight and height as self-reported and not measured might not correspond to the true physiological variables. However, the use of self-reported anthropometric measurements in adults can be used for weight classification purposes [54]. Another limitation of the study is related to the fact that the sampling covered a geographically limited area, so generalizations and extrapolations from the results more broadly should be made cautiously.

## 5. Conclusions

This study demonstrated that, overall, non-professional sportsmen do not have sufficient knowledge on nutrition and that the gym environment does not facilitate the circulation of the correct information on the role of dietary supplements, nor do the coaches in these environments have an appropriate competency level to provide nutritional counseling. The importance of nutrition education has increasingly been recognized, with a consensus that people’s food choices, diet, and physical activity behaviors, influence health and nutritional status. The Italian Dietary Guidelines [55], and public health nutrition messages, stress the importance of physical activity as a determinant of physical well-being and that it constitutes health-promoting behavior. This message risks being undermined if messages inside gyms conflict with recommendations or spread misconceptions about the role of supplements and the risks of searching for shortcuts to improve performance.

Sportsmen, and those with superficial knowledge of nutrition, place great emphasis on the use of dietary supplements; there is a need to counteract this belief and explain that, of all the factors that determine athletic performance, supplements play only a very small role [5]. There is a widespread idea that supplementation with natural products or with vitamins and minerals is safe. This is another aspect that needs to be addressed, considering that high doses of vitamins and minerals are not exempt from risks and, for several of them, an upper level of intake has been established that should not be exceeded [56,57].

The non-professional sports environment is a grey zone in terms of nutritional information in which it is particularly important to provide scientifically correct information about the benefits and risks of using supplements, so that gym users can make informed choices, emphasizing the role of a balanced diet in achieving their specific goals [6].

## Figures and Tables

**Figure 1 nutrients-14-00945-f001:**
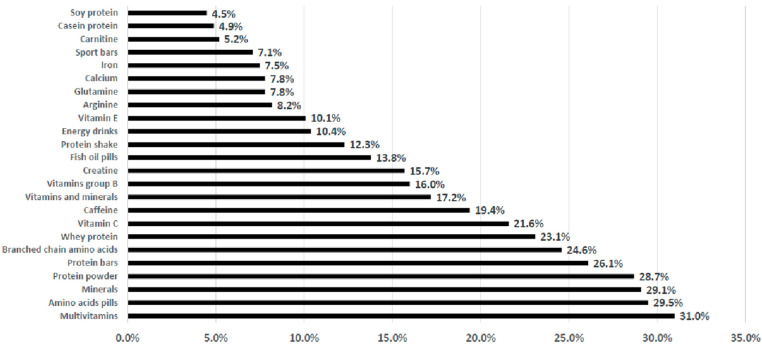
Typology of food supplements used by the sub-sample that reported supplement use (*n* = 268).

**Figure 2 nutrients-14-00945-f002:**
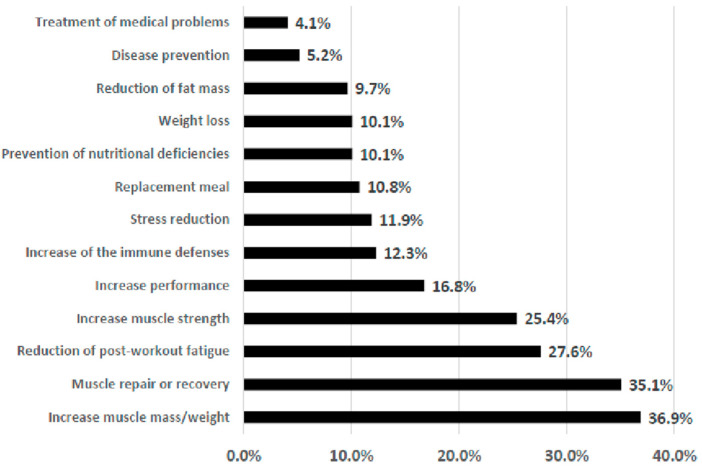
Reasons for supplement usage in the sub-sample that declared use (*n* = 268).

**Figure 3 nutrients-14-00945-f003:**
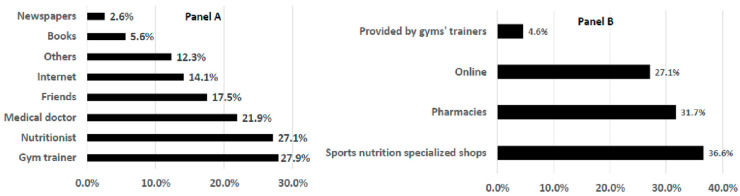
Supplement supply in the sub-sample that reported obtaining them (*n* = 268): Panel A—who suggested the use of supplements? Panel B—where were the supplements bought?

**Table 1 nutrients-14-00945-t001:** Participant characteristics (mean, standard deviation, numbers, percentages).

Characteristics	Total Participants	Males	Females
Sex	581	328 (56.5%)	253 (43.5%)
Age (years)			
Mean ± SD	31.9 ± 8.5	30.5 ± 7.7	33.8 ± 9.1
Classes of age			
20–30	293 (50.4%)	188 (57.3%)	105 (41.5%)
31–40	169 (29.1%)	93 (28.4%)	76 (30.0%)
41–50	119 (20.5%)	47 (14.3%)	72 (28.5%)
BMI (kg/m^2^)			
Mean ± SD	23.3 ± 3.1	24.5 ± 2.7	21.7 ± 3.0
Ponderal status			
Underweight	17 (2.9%)	0 (0%)	17 (7.2%)
Normal	403 (69.4%)	203 (61.9%)	200 (84.7%)
Overweight	145 (25.0%)	114 (34.8)	31 (13.1%)
Obese	16 (2.8%)	11 (3.4)	5 (2.1%)
Education			
Secondary school (Lev1)	42 (7.2%)	25 (7.6%)	17 (6.7%)
Secondary school (Lev2)	307 (52.8%)	194 (59.1%)	113 (44.7%)
Graduate and post-graduate	232 (39.9%)	109 (33.2%)	123 (48.6%)

**Table 2 nutrients-14-00945-t002:** Characterization of the sample in terms of typology and timing of sports and diet attitude.

Question	Options	*n*	%
Period of performing physical activity	Less than 1 month	15	2.6
	1–6 months	63	10.8
	7–12 months	47	8.1
	More than 1 year	456	78.5
Times of training per week	Less than 3 times a week	163	28.1
	3–5 times a week	347	59.7
	More than 5 times a week	71	12.2
Hours of training per day	Less than 1 h per day	76	13.1
	1–2 h per day	445	76.6
	More than 2 h per day	60	10.3
Typology of physical activity	Strength training	415	71.4
	Cardio-fitness activities	61	10.5
	Treadmill	39	6.7
	Water sports	33	5.7
	Team sports	19	3.3
	Wrestling and martial arts	9	1.5
	Yoga	5	0.9
Occurrence of a special diet	Yes	180	31.0
	No	401	69.0
Who recommended the special diet	Nutritionist	87	48.3
	Self-prescribed	44	24.4
	Gym trainers	29	16.1
	No answer	10	5.6
	Medical doctor	7	3.9
	Others	3	1.7

**Table 3 nutrients-14-00945-t003:** Use of different types of food supplements by sample characteristics.

	Gender				Classes of Age					
	F		M		20–30 Years		31–40 Years		41–50 Years	
	*n*	%	*n*	%	*n*	%	*n*	%	*n*	%
General use of supplements	102	38.1	166	61.9 *	131	48.9	77	28.7	60	22.4
Amino acid pills	8	10.1	71	89.9 *	44	55.7	25	31.6	10	12.7
Creatine	7	16.7	35	83.3 *	22	52.4	11	26.2	9	21.4
Glutamine	0	0.0	20	100.0 *	14	70.0	3	15.0	3	15.0
Arginine	2	9.1	20	90.9 *	15	68.2	3	13.6	4	18.2
Branched chain amino acids	10	15.2	56	84.8 *	32	48.5	21	31.8	13	19.7
Casein protein	0	0.0	13	100.0 *	8	61.5	4	30.8	1	7.7
Whey protein	10	16.1	52	83.9 *	36	58.1 *	17	27.4	9	14.5
Protein powder	14	18.2	63	81.8 *	49	63.6	18	23.4	10	13.0
Soy protein	3	25.0	9	75.0 *	7	58.3	3	25.0	2	16.7
Multivitamins	30	36.1	53	63.9	43	51.8	21	25.3	19	22.9
Minerals	39	50.0	39	50.0	30	38.5	24	30.8	24	30.8
Vitamins and minerals	16	34.8	30	65.2	23	50.0	14	30.4	9	19.6
Vitamins group B	21	48.8	22	51.2	19	44.2 *	9	20.9	15	34.9
Vitamin C	21	36.2	37	63.8	34	58.6	13	22.4	11	19.0
Vitamin E	11	40.7	16	59.3	16	59.3	4	14.8	7	25.9
Carnitine	2	14.3	12	85.7 *	7	50.0	4	28.6	3	21.4
Protein shake	7	21.2	26	78.8 *	15	45.5	11	33.3	7	21.2
Protein bars	23	32.9	47	67.1	38	54.3	19	27.1	13	18.6
Energy drinks	6	21.4	22	78.6 *	11	39.3	9	32.1	8	28.6
Caffeine	15	28.8	37	71.2 *	31	59.6	11	21.2	10	19.2
Fish oil pills	12	32.4	25	67.6	14	37.8	10	27.0	13	35.1
Calcium	8	38.1	13	61.9	10	47.6 *	2	9.5	9	42.9
Iron	11	55.0	9	45.0	8	40.0	2	10.0	10	50.0 *
Sport bars	7	36.8	12	63.2	11	57.9	6	31.6	2	10.5

* *p* < 0.05.

**Table 4 nutrients-14-00945-t004:** Reasons for using sport food supplements by sample characteristics.

	Gender				Classes of Age					
	F		M		20–30 Years		31–40 Years		41–50 Years	
	*n*	%	*n*	%	*n*	%	*n*	%	*n*	%
Increase muscle mass/weight	12	12.1	87	87.9 *	64	64.6 *	22	22.2	13	13.1
Increase muscle strength	15	22.1	53	77.9 *	39	57.4	18	26.5	11	16.2
Replacement meal	9	31.0	20	69.0	17	58.6	8	27.6	4	13.8
Muscle repair or recovery	18	19.1	76	80.9 *	51	54.3	26	27.7	17	18.1
Increase performance	8	17.8	37	82.2 *	25	55.6	13	28.9	7	15.6
Disease prevention	2	14.3	12	85.7 *	6	42.9	3	21.4	5	35.7
Prevention of nutritional deficiencies	9	33.3	18	66.7	14	51.9	5	18.5	8	29.6
Weight loss	12	44.4	15	55.6	10	37.0	12	44.4	5	18.5
Treatment of medical problems	11	100.0 *	0	0.0	4	36.4	3	27.3	4	36.4
Increase in immune defenses	17	51.5	16	48.5	13	39.4	11	33.3	9	27.3
Reduction of fat mass	5	19.2	21	80.8	13	50.0	9	34.6	4	15.4
Stress reduction	19	59.4	13	40.6	11	34.4	8	25.0	13	40.6 *
Reduction of post-workout fatigue	18	24.3	56	75.7	36	48.6	22	29.7	16	21.6

* *p* < 0.05.

**Table 5 nutrients-14-00945-t005:** Sport nutrition knowledge assessment of the sample (*n* = 581).

	Total NKS	Macronutrients (NKS1)	Micronutrients (NKS2)	Hydration (NKS3)	Energy Balance (NKS4)	Supplements (NKS5)
Mean (a)	18.8	4.9	3.7	4.2	3.9	2.1
SD	5.4	1.9	1.5	1.5	1.5	1.2
Theoretical max (b)	33	9	6	7	6	5
NK rate (a)/(b)	57.1%	54.2	62.2	60.4	64.2	41.2
Sufficient level (≥60% of correct answers)	47.3%	38.6%	58.2%	43.7%	65.6%	34.3%

**Table 6 nutrients-14-00945-t006:** Rate (%) of NKS sufficiency level by sample characteristics and food supplement behavior.

Characteristics and Behavior	NKS Level			
	Sufficient		Insufficient	
	*n*	%	*n*	%
Sex				
Female	105	38.5	147	48.0
Male	168	61.5 *	159	52.0
Classes of age				
20–30	143	52.0	150	49.0
31–40	78	28.4	91	29.7
41–50	54	19.6	65	21.2
Ponderal status				
Underweight	8	2.9	9	2.9
Normal	184	66.9	219	71.6
Overweight	76	27.6	69	22.5
Obese	7	2.5	9	2.9
Education				
Secondary school (Lev1)	7	2.6	29	9.6
Secondary school (Lev2)	144	52.9	163	53.8
Graduate and post-graduate	121	44.5 *	111	36.6
Use of supplements				
Yes	142	51.6	127	41.5
No	133	48.4	179	58.5 *

* *p* < 0.05.

## Data Availability

The archived data and all the elaboration and analysis generated and used for the presentation of results in this study are fully available on request from the corresponding author.

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
