# Peer review of "Sportsmen’s Attitude towards Dietary Supplements and Nutrition Knowledge: An Investigation in Selected Roman Area Gyms"

_nutrients, 2022, doi:10.3390/nu14050945_

Round 1
Reviewer 1 Report
Finamore et al. have studied the prevalence of dietary supplementation and the nutrition knowledge on sport in Italian gym-goers with a quite nice sample size. The topic is interesting and important, because it may help to understand the reasons for using the supplements, and how the supplement usage and nutrition knowledge may be linked.
I have some comments, that perhaps may help to improve the paper before it can be published. First, I think that the English language should be revised throughout the article, and I have not emphasized all the problematic sentences one by one in my report. It should also be checked, that point (.) is used as a decimal point, not comma (,).
Abstract: Should it be mentioned already here, what was considered as sufficient level of NKS?
Line 62: Is this the correct way to write references, or should the numbers be written inside the same square brackets?
Lines 138-139: A bit unclear sentence, please check and rephrase.
Line 167: How you chose the 33 questions that you used in your study to asses the nutrition knowldege, and is it ok to use the same standard 60 percent cut point to define adequate level of nutrition knowledge, when using less questions than the original questionnaire?
Line 196: What kind of pathologies you are referring to, and what pathologies did those 8.6% have?
Line 199: What does the "numbers" mean in the caption?
Table 1: Should the characteristic be presented in men and women separately?
Lines 210-212: I am afraid I don't understand this sentence.
Line 221: Creatine, not creatinine
Lines 220-227 and 258-259 and 280-281: I think these could be written a bit more clearly. I'm not sure if it is now clear where these numbers are compared to and what difference do the p-values represent.
Line 299: Can you be sure that these respondents were obese?
Lines 300-302: Please check. Maybe some rephrasing needed.
Reviewer 2 Report
I understand that English is not your native language, therefore, there is the necessity for extensive English corrections.
- You can use the terms of Macro and Micronutrients. Then talk about the supplements
and the kind of them.
- Please rephrase the sentence.
- Change “Users” with “participants”
- Please rephrase the sentence.
58-59. Check the sentence I cannot understand the meaning.
- Explain the abbreviation “EU”.
- Legal ergogenic supplements are included Carbohydrates and Sodium Nitrate.
- Please rephrase the sentence.
- Did you use any inclusion criteria for the participants’ selection?
- Is there literature in which you can regard this choice?
- It is preferable to describe the questionnaire in a separate section.
- Are there correct or false answers?
- with males … Please, rephrase the sentence
- The symbols’ explanation must be written at the end of the table. Please, change it wherever is necessary.
- Correct the sentence grammatically.
- To “record” not “investigate”.
- Avoid 1st plural. Use passive voice.
354 – 356. Please rewrite the sentence.
- Change “in our”
Round 2
Reviewer 2 Report
176. "who agreed" not "that"
273. Put "." not "," 78.6%
282. Explain that this means a statistically significant difference. Change it wherever it is necessary.
Author Response
We thank you for this second-round review. In the re-submitted manuscript, you could find all the changes (in green) carried out according to your suggestions.
